# AXL and MET Tyrosine Kinase Receptors Co-Expression as a Potential Therapeutic Target in Malignant Pleural Mesothelioma

**DOI:** 10.3390/jpm12121993

**Published:** 2022-12-02

**Authors:** Federica Zito Marino, Carminia Maria Della Corte, Vincenza Ciaramella, Stefania Erra, Andrea Ronchi, Alfonso Fiorelli, Giovanni Vicidomini, Mario Santini, Giosuè Scognamiglio, Floriana Morgillo, Fortunato Ciardiello, Renato Franco, Marina Accardo

**Affiliations:** 1Pathology Unit, University of Campania “Luigi Vanvitelli”, 80138 Naples, Italy; 2Department of Precision Medicine “F. Magrassi e A. Lanzara”, Institute of Medical Oncology, 80138 Naples, Italy; 3Pathology Unit, ASL AL, 15033 Casale Monferrato, Italy; 4Translational Medical and Surgical Science, University of Campania “Luigi Vanvitelli”, 80138 Naples, Italy; 5Pathology Unit, Istituto Nazionale Tumori IRCCS “Fondazione G. Pascale”, 80131 Naples, Italy

**Keywords:** malignant pleural mesothelioma (MPM), AXL, MET, targeted tyrosine kinase inhibitor (TKIs)

## Abstract

Malignant pleural mesothelioma (MPM) is a highly lethal malignancy that unfortunately cannot benefit from molecularly targeted therapies. Although previous results showed the pivotal role of various receptor tyrosine kinases (RTKs) in MPM tumorigenesis, the treatment with a single inhibitor targeting one specific RTK has been shown to be ineffective in MPM patients. The main aim of the present study was to investigate the potential role of AXL and MET receptors in MPM and the possible efficacy of treatment with AXL and MET multitarget inhibitors. Immunohistochemical and FISH analyses were performed in a wide series of formalin-fixed paraffin-embedded MPM samples to detect the expression of two receptors and the potential gene amplification. In vitro studies were performed to evaluate putative correlations between the target’s expression and the cell sensitivity to AXL-MET multitarget inhibitors. In our series, 10.4% of cases showed a co-expression of AXL and MET, regardless of their ligand expression, and the gene amplification. Furthermore, our in vitro results suggest that the concomitant pharmacological inhibition of AXL and MET may affect the proliferative and aggressiveness of MPM cells. In conclusion, the subset of MPM patients with AXL-MET co-activation could benefit from treatment with specific multitarget inhibitors.

## 1. Introduction

Malignant pleural mesothelioma (MPM) is a highly aggressive and rare tumor arising from the pleura [1]. MPM is difficult to treat and commonly associated with asbestos exposure, which is its main risk factor [2]. MPM incidence is increasing worldwide, and a peak is expected in the coming decades. However, the exact time of this peak will vary among countries, owing to differences in the timing of the reduction in or the prohibition of asbestos use [3]. Histologically, three main MPM subtypes are distinguished: the epithelioid, the sarcomatoid and the biphasic, with the latter combining features of the other two histotypes [4]. The epithelioid is the most common (60% or more) histological subtype and the less aggressive; the sarcomatoid is the rarest subtype (<10%) but the most aggressive; finally, the biphasic histotype (10–15% of cases) with an intermediate prognosis between the other two subtypes [5,6]. The long-term survival rate of MPM is poor, ranging approximately from 14 to 15 months despite the treatment. Therapeutic options include multi-modality approach based on chemotherapy, surgical resection, and thoracic radiation, lacking any personalized treatment strategies [7]. In recent years, extensive research has focused on the identification of prognostic and predictive markers, however, distinct from other cancers, no target therapies have been currently approved for MPM. The genetic landscape of MPM is characterized by several genetic alterations including the deregulated activation of several signaling pathways, particularly the canonical receptor tyrosine kinase pathways [8,9]. Hmeljak et al. proposed a new integrated molecular classification of MPM, showing for the first-time subgroups of MPM from a biological point of view [10]. Among all the proposed targets emerged, the activation of various receptor tyrosine kinases (RTKs) plays a central role in MPM pathogenesis, leading to the oncogenic progression of non-neoplastic mesothelial progenitor to malignant cells. In vitro studies showed that drugs targeting RTKs might be candidate inhibitors in the treatment of MPM based on the activation of several kinase signaling pathways in this cancer [11]. Despite the proven RTK activation in mesothelioma and encouraging in vitro results with RTK inhibitors, clinical trials have shown no relevant clinical activity of any single targeted tyrosine kinase inhibitor (TKIs) in MPM patients [12]. The limited clinical success of a single agent could be justified by the co-activation of multiple RTKs resulting in mesothelioma cell proliferation and survival. Several studies demonstrated the activation of multiple RTKs in MPM, including EGFR, MET, PDGFRA, PDGFRB, and IGF1R [11]. MET is a RTK involved in cell growth, replication, and motility, that could play a role in tumorigenesis of several human cancers through multiple mechanisms, including altered regulation, genetic mutations, and upregulation of its ligand hepatocyte growth factor (HGF) [13]. MET overexpression has been reported in approximately 74–100% of MPMs, although the molecular mechanisms of its upregulation remain incompletely understood [14,15]. MET gene mutations have been identified in 3–16% of MPMs, resulting in a variable clinical value [11,14]. MET amplification in parallel with MET over-expression has been described in non-small cell lung cancer and gastrointestinal tumors, while it represents a rare event in MPM with a frequency less than 1% [16]. Previous in vitro studies showed the efficacy of MET inhibition by small molecule inhibitor or RNAi knockdown in MPM cell lines, resulting in the decreased phosphorylation of MET and the cell cycle arrest [12]. AXL is a member of the TAM RTK subfamily, implicated in cell survival, epithelial-to-mesenchymal transition, and drug resistance. AXL can be activated through different mechanisms, including ligand-independent dimerization and ligand-dependent dimerization especially driven by Gas-6 its major ligand. AXL overexpression has been associated with adverse prognosis in several neoplasms such as hepatocellular carcinoma, esophageal carcinoma, pancreatic carcinoma, thyroid carcinoma, and colon carcinoma [17]. Pinato et al. have validated AXL overexpression in a large series of MPM confirming an independent prognostic value of the receptor expression. Moreover, the prognostic power of AXL expression could be stronger than that of commonly used prognostic factors including tumor stage and EORTC prognostic score [18]. In vitro studies have previously documented that AXL inhibition can suppress the proliferation and the invasion of mesothelioma cells suggesting its potential role as therapeutic target in MPM [12,18]. The main aim of the present study was to investigate the potential role of AXL and MET receptors in MPM. First, we analyzed AXL and MET overexpression in a wide series of MPM patients through immunohistochemical analyses, and moreover the amplification of both two genes was investigated by FISH. Then, a second objective was to assess in vitro efficacy of treatment with AXL and MET multitargets inhibitors, to evaluate putative correlations between the target’s expression and the cell’s drug sensitivity.

## 2. Materials and Methods

### 2.1. Patients Cohort

A series of 72 MPM tumors from patients undergoing surgical resection or biopsies at the Università degli Sudi della Campania Luigi Vanvitelli and the Ospedale Santo Spirito Casale Monferrato were collected. The following clinical and pathological parameters were evaluated for the patient’s cohort: patient age at initial diagnosis, smoking habits, histologic grade, tumor stage, tumor recurrence or distant metastasis, asbestos exposure. Sections of 4 μm thickness from each block were obtained and were stained with hematoxylin-eosin. All 72 cases of MPM were reviewed by two expert pulmonary pathologists. The disease stage was defined according to the eighth TNM classification for malignant pleural mesothelioma [19].

### 2.2. Immunohistochemistry

Immunohistochemistry analysis was performed to evaluate AXL, Gas6, MET and HGR expression. Immunohistochemical staining was carried out on TMAs slides using the following specific antibodies: anti-AXL (Human Axl Antibody Antigen Affinity-purified Polyclonal Goat IgG;R&D System); dilution 1:20, incubation over-night; anti-Gas6 (Human Gas6 Antibody Antigen Affinity-purified Polyclonal Goat IgG; R&D System); dilution 1:50, incubation over-night; anti-Met (Human MET Antibody Antigen Affinity-purified Monoclonal Mouse; Cell Signaling) dilution 1:100, incubation 1 h; anti-HGF (Abcam ab216623) dilution 1:100, incubation 1 h. Paraffin slides were deparaffinized in xylene and rehydrated through graded alcohols. Antigen retrieval was performed with slides heated in EDTA buffer (pH 9.0) in a bath for 20 min at 97 °C. After antigen retrieval, the slides were allowed to cool. The slides were rinsed with TBS and the endogenous peroxidase was inactivated with 3% hydrogen peroxide. After protein block (BSA 5% in PBS 1×), the slides were incubated with specific antibodies according to the condition described above. The sections were incubated with biotinylated anti-rabbit antibody for 40 min at room temperature. Immunoreactivity was visualized by means of avidin-biotin-peroxidase complex kit reagents (Novocastra, Newcastle, UK) as the chromogenic substrate. Finally, sections were weakly counterstained with hematoxylin and mounted. IHC scoring was based on the cytoplasmic and/or membrane-staining intensity, as follows: no staining or weak staining in <10% of tumor cells, score 0; weak staining in >10% of tumor cells, score 1+; moderate staining in >10% of tumor cells, score 2+; strong staining in >10% of tumor cells, score 3+.

### 2.3. Fluorescence In Situ Hybridization

Fluorescence in situ hybridization (FISH) analysis was performed on unstained 5 μm thick TMAs slides to detect AXL and MET amplification. The deparaffinization of sections was carried out with two 10 min immersions in bioclear, followed by three 3 min immersions in ethanol 100, 70 and 50%. The slides were rinsed in distilled water and immersed in Vysis pretreatment solution (1Msodium thiocyanate) at 80 °C for 10 min, and in protease solution (previously warmed to 37 °C) for 10 min, washed with purified water, air-dried, and dehydrated in ascending grades of alcohol. The used probes are commercially available: Abnova AXL/CEN19q FISH Dual Color Probe consisting of a TexRed fluorochrome direct labeled SPEC AXL probe hybridizing to the AXL gene in the chromosomal region 19q13.2 and FITC fluorochrome direct labeled SPEC 19q12 probe; ZytoLight SPEC MET/CEN 7 Dual Color Probe consisting of one orange fluorochrome direct labeled CEN 7 probe specific for the alpha satellite centromeric region of chromosome 7 (D7Z1) and a green fluorochrome direct labeled SPEC MET probe specific for the MET gene located at 7q31.2. Denaturation and hybridization of the tissue sections were performed using the Thermobrite system (Abbott Molecular Inc., Des Plaines, IL, USA): 75 °C for 5 min for the denaturation process and 37 °C for 17 h for the hybridization of the probes. The slides were then washed with 0.4× saline-sodium citrate (SSC) solution at 70 °C for 2 min and 2× SSC at room temperature for 3–5 min. Lastly, 10 μL of DAPI was applied on the slides. The fluorescence signals were evaluated under epifluorescence microscope (Olympus) and the image acquisition was carried out by CCD microscopy camera. The fluorescence signals were evaluated under epifluorescence microscope. The signals were counted in 20 tumor nuclei in a minimum of two areas to determine the MET/CEN-7 and AXL/CEN19 ratio. The cells of the mesothelium were used as normal counterpart for FISH evaluation. The FISH scoring was defined as follows: ratio ≤ 2.0: normal cell; ratio > 2.0: gene amplification.

### 2.4. In Vitro Studies

#### 2.4.1. Cell Lines

In our study, we selected a wide panel of several human MPM cell lines according to different histological type of original tumor, including epithelioid type (NCI-H2052, NCI-H2452), sarcomatoid type (NCI-H28), and a biphasic type (MSTO-211H). Cell lines were maintained in Dulbecco’s modified Eagle’s medium (DMEM) supplemented with 10% fetal bovine serum (FBS; Life Technologies, Gaithersburg, MD, USA) and 1% antibiotics/antimycotics (Life Technologies, Gaithersburg, MD, USA).

#### 2.4.2. Western Blot Analysis

Protein lysates were obtained by homogenization in RIPA lysis buffer (Sigma-Aldrich, St. Louis, MO, USA) with protease and phosphatase inhibitors cocktail (Hoffmann-La Roche). Protein extracts were quantified by using the Bradford assay (Bio-Rad, Hercules, CA, USA) and equal amounts of total protein (40 μg/lane) were separated by 4–15% gradient mini precast TGX gel (Bio-Rad) before transferring to nitrocellulose by standard western conditions, blocked in BSA solution and primary antibodies (1:1000 in BSA solution; incubated overnight at 4 °C). The secondary antibody (1:3000 in 5% milk/TBS/Tween20 solution) was incubated at RT for 1 h before detection. Immunocomplexes were detected with the enhanced chemiluminescence kit ECL plus, by Thermo Fisher Scientific (Rockford, IL, USA) using the ChemiDoc (Bio-Rad). Values were normalized to α-tubulin. Each experiment was performed in duplicate.

#### 2.4.3. Cell Proliferation Assays

Cancer cells were seeded in 96-well plates and treated with different doses of indicated drug for 72 h as single agent or in combination. Cell proliferation was measured with the MTT assay, that was performed following the manufacturer’s instructions as previously described. IC50 was determined by interpolation from the dose-response curves. The results represent the median of three separate experiments, each performed in quadruplicate.

#### 2.4.4. Colony Forming Assays

Cells were seeded on 6-well tissue culture dishes at 300 cells/well and were treated with indicated drugs at IC50 doses. All of the experiments were performed in triplicate and untreated cells were used as control. Cells were maintained for 7 days at which point they were fixed with 4% paraphormaldeid, stained with crystal violet and colonies counted using the GelCount (Oxford Optronix, UK).

### 2.5. MET and AXL Co-Expression

In order to verify the co-expression of AXL and MET, a double immunofluorescence assay was performed. H2452 cells were fixed with 4% formaldehyde for 15 min at room temperature. FFPE 4 microns sections were deparaffinated and rehydratated, followed by antigen retrieval. Both H2452 cells and FFPE tissues were incubated overnight at 4 °C with a mixture of anti-AXL (Human AXL Antibody Antigen Affinity-purified Polyclonal Goat IgG; R&D System); dilution 1:20 and anti-MET (Human MET Antibody Antigen Affinity-purified Monoclonal Mouse; Cell Signaling) dilution 1:100. The incubation of 1 h with rabbit anti-goat gG H&L (Alexa Fluor 594) (Abcam, ab150148) and rabbit Anti-Mouse IgG H&L (Alexa Fluor^®^ 488) (Abcam, ab150117) secondary antibodies was performed. The sections were mounted using a medium with 4′,6-diamidino-2-phenylindole (DAPI) to visualize nuclear details (ab104139). Immunofluorescence-labeled sections were imaged using a fluorescence microscope (Leica, DM600B) equipped with the appropriate filter sets. H2452 cells and the tissue sections were acquired to 100× and 63× magnification, respectively.

### 2.6. Statistical Analysis

The Pearson Χ-square test was performed to determine whether a relationship exists between the AXL and MET proteins expression and the clinical-pathological features. *p* < 0.05 (2-sided) was evaluated as statistically significant. Data analysis and summarization were conducted using SPSS 20.0 for Mac (SPSS Inc., Chicago, IL, USA).

## 3. Results

### 3.1. Clinical and Pathological Characteristics of Patients

Clinical and pathological features of 72 MPM patients are shown in Table 1. Among the patients analyzed, 61 out of 72 (84%) were male patients. The mean age was 73 years (range 44–82). Among all patients, 44.4% were stage I, 33.3% stage II, 13.8% stage III, and 8.3% stage IV. The majority of the patients (79.1%) enrolled in our study had previously been exposed to asbestos. The majority of the MPMs analyzed were epithelial (76.3%) followed by the sarcomatoid (12.5%) and biphasic (11.1%) (Table 1).

### 3.2. AXL IHC and FISH Results

Overall, AXL expression was found in 11 out of 72 cases; all clinical and pathological features are summarized in Table 2 AXL IHC membrane and cytoplasmatic staining were interpreted in accordance with the criteria described above. The representative results are showed in Figure 1d–f. Among eleven IHC positive cases, seven cases were score 3, two cases score 2 and two cases score 1. No correlation was found between AXL and Gas6 IHC expression, suggesting a possible ligand-independent expression of the receptor.

AXL gene amplification was found in 10 out of 72 cases analysed. AXL gene amplification was interpreted in accordance with the criteria described above, representative results are showed in Figure 2a. Among ten cases with AXL gene amplification, only four cases showed AXL expression (3 cases were score 3 and 1 case was score 1), suggesting no close association between the activation of the receptor and the gene amplification.

### 3.3. MET IHC and FISH Results

Overall, MET expression was found in 15 out of 72 cases, and all clinical and pathological features are summarized in Table 2. MET IHC staining were interpreted in accordance with the criteria described above, representative results are showed in Figure 1g–i. Among fifteen IHC positive cases, ten cases were score 3, four cases score 2 and one case was score 1. No correlation was found between MET and HGF IHC expression, suggesting a possible ligand-independent expression of the receptor.

MET gene amplification was found only in one out of seventy-two cases analysed. MET gene amplification was interpreted in accordance with the criteria described above, and representative results are showed in Figure 2c. No cases that showed MET expression harbouring MET gene amplification, suggesting neither association between the activation of the receptor and the gene amplification.

### 3.4. AXL and MET Correlation

In our series, seven out of seventy-two cases analyzed showed a co-expression of AXL and MET, regardless of their ligand expression, and genes amplification (Table 2). The immunofluorescence confirmed the co-expression of the two biomarkers in neoplastic tissues. In particular, a cytoplasmic co-expression of both markers was observed in the same cells (Appendix A).

MPM harboring AXL and MET co-expression showed a statistically significant correlation with epithelioid histotype (*p* < 0.05).

### 3.5. In Vitro Studies

In order to explore the potential therapeutic value of the expression of MET and AXL in mesothelioma, we evaluated the expression level of these protein in four human mesothelioma cell lines, representative of different mesothelioma subtypes: H2452, H2052, H28 and MSTO, epithelial, sarcomatoid and biphasic mesothelioma, respectively. The immunofluorescence analysis demonstrated the co-expression of the two biomarkers in H2452 cells (Appendix A).

Western blot analysis revealed no significant difference in expression levels of total and phosphorylated isoforms of each protein in all cell lines used, with the exception of H2452 cell lines with lower expression of phospho-AXL (Figure 3). We selected two multi-target inhibitors to test the in-vitro anti-tumor activity of AXL and MET inhibition: foretinib, exerting antitumor activity via inhibition of MET, AXL and VEGF receptor 2 and cabozantinib, with activity against a broad range of targets, including MET, RET, AXL, VEGFR2, FLT3, and c-KIT. A dose-dependent inhibition of cell viability was examined after 72 h treatment of human mesothelioma cell lines with increasing doses of the drugs, and we observed a decrease in cell proliferation with IC50 value between 0.5 and 1 µM for foretinib and between 1 µM and 2 µM for cabozantinib (Figure 3). Similar effects were detected for colony forming abilities; treatment with IC50 dose of indicated drugs induced reduction of this ability of all cell lines to at least 30% (Figure 3).

## 4. Discussion

MPM is a notoriously chemotherapy-resistant neoplasm with no target therapeutic options. To date, unfortunately no target therapy has been recommended for MPM patients [20]. Tyrosine kinase proteins are key regulatory elements of proliferation and survival in many cancers, their activation is essential in the development of mesothelioma from the starting point of a non-neoplastic mesothelial progenitor cell. In this context, the development of new alternative therapeutic strategies for the treatment of MPM asre urgently required, the multitarget RTK inhibitors might represent a glimmer in this highly lethal disease [14]. In MPM, constitutive activation of phosphoinositide-3 kinase/Akt (PI3/AKT) is frequent and contributes to the malignant phenotype, also supporting a role for signaling from RTKs through PI3/AKT in this cancer [21]. The co-expression of RTKs is already described in MPM; particularly EGFR and MET proteins are co-activated in MPM cell lines, whereas EGFR and PDGFRB co-expression was described in human tissues of MPM [14]. Previous data have shown the co-activation of the receptor tyrosine kinases (RTKs) in MPM, suggesting that multitargets TKs might represent therapeutic targets in this highly lethal disease [11]. Our results in FFPE specimens showed that AXL and MET can be simultaneously expressed in MPM, suggesting the possible cooperation of these RTKs in the pathogenesis of this neoplasia. Brevet et al. showed the co-activation of MET and EGFR in most cases of MPM, and they described, for the first time, the codependence of the EGFR and MET signaling in MPM [11]. Moreover, they find that different RTKs can contribute to feedback activation of AKT caused by mTOR inhibition in MPM, imparting sensitivity to combinations of rapamycin with different RTK inhibitors [11]. These observations may explain the disappointing results obtained with single-agent RTK inhibitors. To date, single-agent RTK specific inhibition is clinically ineffective in patients with MPM, despite other cancer types. The concomitant activation of several RTKs may allow tumor cells to acquire resistance to TKI monotherapies, and combination of RTKs inhibitors may, therefore, be necessary in MPM. Previous data showed that pharmacologic inhibitors of MET or AXL alone do not show broad activity against MM cell lines [14]. Notably, AXL expression, overall, was even stronger in mesothelioma than in various other cancer types that have been reported to feature high levels of AXL expression. AXL and MET share several structural features and they are similarly implicated in cell migration, invasion, metastasis, and drug resistance. Previous published studies showed biochemical interplay of the pathways activated by ALK and MET [22]. Recently, a role for AXL in the regulation of MET and EGFR signaling was described, indicating that AXL may also function as an important factor mediating resistance against specific TKIs. The AXL and MET co-expression and our in vitro results suggest a possible use of TKI multitargets inhibitors in the treatment of this tumors. New multitargets inhibitors, particularly specific for AXL and MET, may be candidate drugs in MPM. For example, cabozantinib, a specific VEGFR2, MET, and AXL multitargets inhibitor, was investigated in a wide range of human cancers and it is currently approved for the treatment of metastatic patients with renal cell carcinomas [23]. Furthermore, our in vitro results showed that the concomitant inhibition of AXL and MET in vitro can affect the proliferation and survival of MPM cancer cells, regardless of the drugs used.

## 5. Conclusions

In conclusion, the co-activation and the crosstalk between different RTKs signaling in MPM could represent a rationale for a combination targeting of kinase signaling pathways drugs. In this context, MET and AXL receptors could play a critical role in MPM and the inhibition of signaling pathways could be a novel therapeutic approach in the treatment of this neoplasia, especially in chemotherapy-resistant patients. Further studies are needed to better explore the subset of MPM that may benefit from the targeting of AXL and MET.

## Figures and Tables

**Figure 1 jpm-12-01993-f001:**
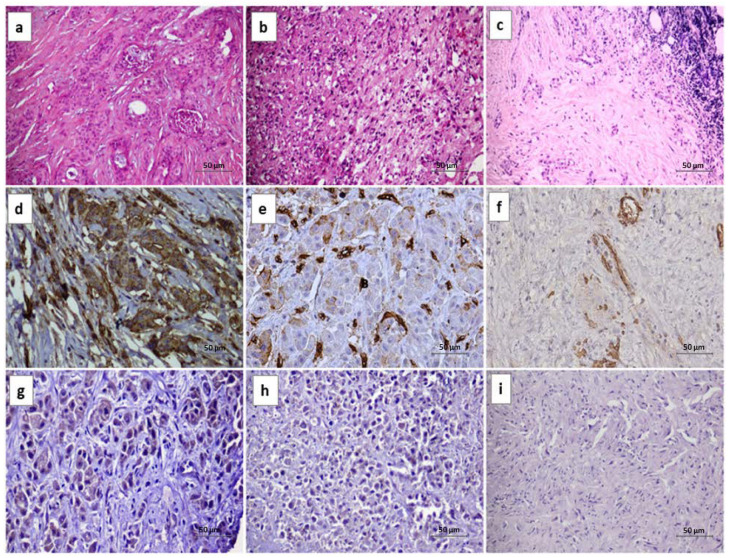
Representative Hematoxylin and Eosin staining and IHC results (original magnification 40×). (**a**) epithelial malignant pleural mesothelioma; (**b**) sarcomatoid malignant pleural mesothelioma; (**c**) biphasic malignant pleural mesothelioma; (**d**) AXL IHC positive staining score 3+; (**e**) AXL IHC positive staining score 2+; (**f**) AXL IHC positive staining score 1+; (**g**) MET IHC positive staining score 3+; (**h**) MET IHC positive staining score 2+; (**i**) MET IHC positive staining score 1+.

**Figure 2 jpm-12-01993-f002:**
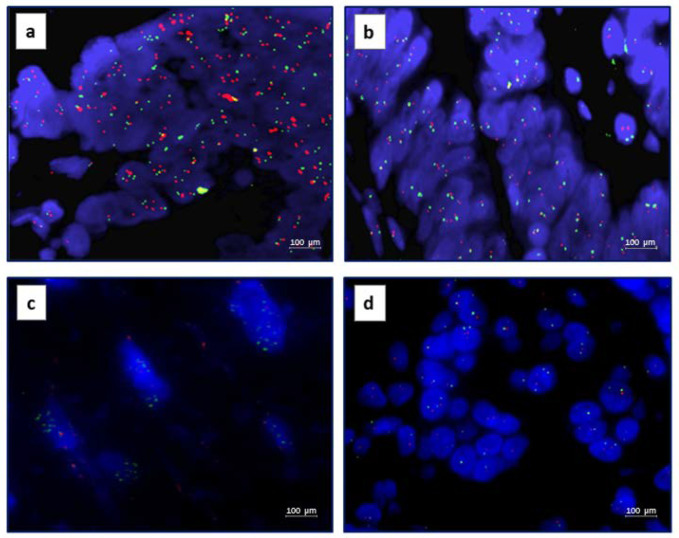
Representative FISH results (original magnification 100×). (**a**) AXL gene amplification; (**b**) AXL gene no amplificated; (**c**) MET gene amplification; (**d**) MET gene no amplificated.

**Figure 3 jpm-12-01993-f003:**
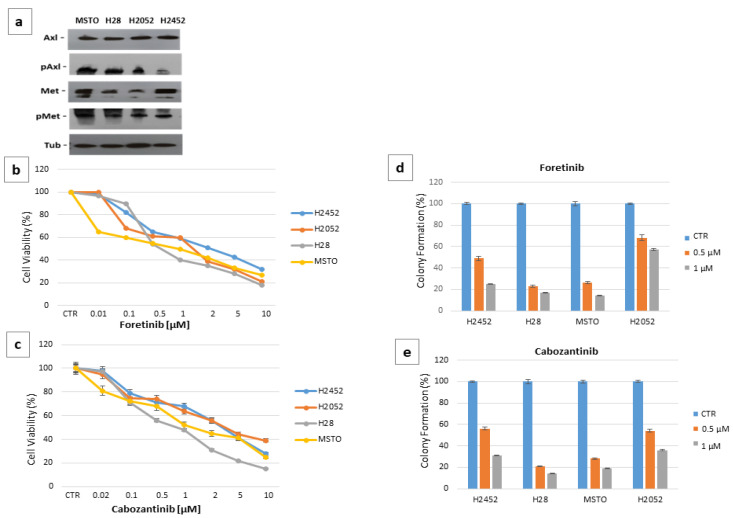
Representative in vitro results of the treatment of human mesothelioma cell lines with increasing doses of the AXL-MET specific inhibitors. (**a**) Western blot analysis for AXL, phospho-AXL, MET, phospho-MET were performed on protein lysates from MISTO, H28, H2052 and H2452 mesothelioma cell lines. α-Tubulin was included as a loading control. (**b**) effect of foretinb on the proliferation of mesothelioma cell lines, 72 h treatment; (**c**) effect of cabozantinib on the proliferation of mesothelioma cell lines, 72 h treatment; (**d**) colony forming abilities of mesothelioma cell lines after the treatment with foretinib; (**e**) colony forming abilities of mesothelioma cell lines after the treatment with cabozantinib. H2452: epitheliod mesothelioma cell lines, H28: sarcomatoid mesothelioma cell lines, MSTO: biphasic mesothelioma cell lines, H205: epithelioid mesothelioma cell lines.

**Table 1 jpm-12-01993-t001:** Clinical and pathological features of patients.

Characteristics	N. (%)
** All cases **	72
** Age, years **	
>73 y	41 (57%)
<73 y	31(43%)
** Gender **	
male	61 (84%)
female	11 (16%)
** Histotype **	
Epithelial	55(76.3%)
Sarcomatoid	9(12.5%)
Biphasic	8(11.1%)
** Disease stage **	
I	32(44.4%)
II	24(33.3%)
III	10(13.8%)
IV	6(8.3%)
** Asbesto exsposure **	
yes	57(79.1%)
no	7(9.7%)
NA	8(11.1%)

**Table 2 jpm-12-01993-t002:** IHC and FISH results of AXL/MET and the clinical-pathological features of the series analyzed.

Characteristics n. (%)	AXL IHC n. (%)	MET IHC n. (%)	AXL-MET Co-Expression n. (%)	FISH AXL n. (%)	FISH MET n. (%)
-	+	-	+	-	+	NA	A	NA	A
** All cases **	72	61 (84.7)	11 (15.3)	57 (79.2)	15 (20.8)	65 (90.3)	7 (9.7)	62 (86.1)	10 (13.9)	71 (98.6)	1 (1.4)
** Age, years **											
>73 y	41 (57%)	31 (50.8)	10 (90.9)	32 (56.1)	9 (60.0)	34 (52.3)	7 (100)	35 (56.5)	6 (60.0)	40 (56.3)	1 (100)
<73 y	31 (43%)	30 (49.2)	1 (9.1)	25 (43.9)	6 (40.0)	31 (47.7)	0 (0)	27 (43.5)	4 (40.0)	31 (43.7)	0 (0)
** Gender **											
male	61 (84%)	50 (82.0)	11 (100)	47 (82.4)	14 (93.4)	54 (83.0)	7 (100)	52 (83.9)	9 (90.0)	60 (84.5)	1 (100)
female	11 (16%)	11 (18.0)	0 (0)	10 (17.6)	1 (6.6)	11 (17.0)	0 (0)	10 (16.1)	1 (10.0)	11 (15.5)	0 (0)
** Histotype **											
Epithelial	55 (76.3%)	44 (72.1)	11 (100)	42 (73.7)	13 (86.6)	48 (73.9)	7 (100)	47 (75.8)	8 (80.0)	54 (76.0)	1 (100)
Sarcomatoid	9 (12.5%)	9 (14.8)	0 (0)	9 (15.8)	0 (0)	9 (13.8)	0 (0)	9 (14.5)	0 (0)	9 (12.7)	0 (0)
Biphasic	8 (11.1%)	8 (13.1)	0 (0)	6 (10.5)	2 (13.4)	8 (12.3)	0 (0)	6 (9.7)	2 (20.0)	8 (11.3)	0 (0)
** Disease stage **											
I	32 (44.4%)	23 (37.7)	9 (81.8)	22 (38.6)	10 (66.6)	26 (40.0)	6 (85.7)	23 (37.1)	9 (90.0)	31 (43.7)	1 (100)
II	24 (33.3%)	23 (37.7)	1 (9.1)	21 (36.9)	3 (20.0)	23 (35.4)	1 (14.3)	23 (37.1)	1 (10.0)	24 (33.8)	0 (0)
III	10 (13.8%)	9 (14.8)	1 (9.1)	8 (14)	2 (13.4)	10 (15.4)	0 (0)	10 (16.1)	0 (0)	10 (14.0)	0 (0)
IV	6 (8.3%)	6 (9.8)	0 (0)	6 (10.5)	0 (0)	6 (9.2)	0 (0)	6 (9.7)	0 (0)	6 (8.5)	0 (0)
** Asbesto exsposure **											
yes	57 (79.1%)	48 (78.7)	9 (81,8)	46 (80.7)	11 (73.4)	51 (78.6)	6 (85.7)	49 (79.0)	8 (80.0)	57 (80.4)	0 (0)
no	7 (9.7%)	7 (11.5)	0 (0)	6 (10.5)	1 (6.6)	7 (10.7)	0 (0)	7 (11.3)	0 (0)	7 (9.8)	0 (0)
NA	8 (11.1%)	6 (9.8)	2 (18.2)	5 (8.8)	3 (20.0)	7 (10.7)	1 (14.3)	6 (9.7)	2 (20.0)	7 (9.8)	1 (100)

NA: not gene amplification; A: gene amplification.

## Data Availability

Not applicable.

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
