# Peer review of "AXL and MET Tyrosine Kinase Receptors Co-Expression as a Potential Therapeutic Target in Malignant Pleural Mesothelioma"

_jpm, 2022, doi:10.3390/jpm12121993_

Round 1
Reviewer 1 Report
In the manuscript entitled "AXL and MET tyrosine kinase receptors co-expression as potential therapeutic target in malignant pleural mesothelioma", the authors tried to investigate the expression and therapeutic value of AXL and MET in malignant pleural mesothelioma. Date indicated that AXL and MET co-expressed in a subset of malignant pleural mesothelioma patients. Treatment of either foretinib or cabozantinib can reduce cell viability and colony formation of the four mesothelioma cell lines. Overall, there is clinical value of this study. However, some key experiments / results are missing.
1) Regarding ethical issue and the use of patients' samples in this study, can you clarify if the patients sign Informed Consent?
2) The Abstract mentioned "our in vitro results showed that the concomitant inhibition of AXL and MET in vitro can affect the proliferative and invasiveness ability of cancer cells". However, no data were shown in the results part. This is key data to the manuscript. The authors should add the results/data to support their conclusion.
3) Where are the apoptosis data?
4) Add immunostaining results to show the co-expression of AXL and MET in mesothelioma patients' samples.
5) Results mentioned "Western blot analysis revealed no significant difference in expression levels of total and phosphorylated isoforms of each protein in all cell lines used, with exception of H2452 cell lines with lower expression of phospho-AXL." Add the results.
6) Show the coexpression of AXL and MET in the cell lines.
7) Add the experimental procedure of Western blot to the "Materials and Methods".
8) Add scale bars to all the images.
9) Make the figure formats the same of Figure 3a and 3b.
Make sure the writing is correct, eg, it should be "0.5" other than"0,5".
10) Why are the error bar of the CTR in Figure 3c and 3d are the same?The data or results should be wrong.
Author Response
1) Regarding ethical issue and the use of patients' samples in this study, can you clarify if the patients sign Informed Consent?
This is a retrospective study that was approved by local EC (N.280 of 16 May 2020). We inserted this information in the main text in the section “Institutional Review Board Statement”.
2) The Abstract mentioned "our in vitro results showed that the concomitant inhibition of AXL and MET in vitro can affect the proliferative and invasiveness ability of cancer cells". However, no data were shown in the results part. This is key data to the manuscript. The authors should add the results/data to support their conclusion.
We thank the reviewer for the comments. We were referring to preclinical results presented in figure 3. We modified the section “Materials and methods”, the legend of Figure 3 and the sentence in the abstract in order to better represent the results highlighted in the text.
3) Where are the apoptosis data?
We apologize for the mistake, we have incorrectly entered in the Materials and Methods the section “Assessment of apoptosis”. The apoptosis experiments were not performed in our study, thus we have eliminated the paragraph 2.4.4 from the main text.
4) Add immunostaining results to show the co-expression of AXL and MET in mesothelioma patients' samples.
As correctly suggested by the referee, the immunostaining results of the co-expression of AXL and MET would be a real added value to the quality of our manuscript. Since both receptors have cytoplasmic staining, therefore the double staining using two different chromogens (for example DAB and fast red) has technical and interpretative problems. Unfortunately, we are unable to provide the co-staining images of the two receptors since they have the same cellular localization.
5) Results mentioned "Western blot analysis revealed no significant difference in expression levels of total and phosphorylated isoforms of each protein in all cell lines used, with exception of H2452 cell lines with lower expression of phospho-AXL." Add the results.
We have performed the western blot analysis to evaluate the expression levels of total and phosphorylated isoforms of AXL and MET, however we had not inserted the related images in the main text. As correctly suggested by the referee, we added the western blot results in Figure 3.
6) Show the coexpression of AXL and MET in the cell lines.
We have performed the western blot analysis to evaluate the expression levels of total and phosphorylated isoforms of AXL and MET, however we had not inserted the related images in the main text. As correctly suggested by the referee, we added the western blot results in Figure 3, showing the protein expression of AXL and MET receptors, along with the respective phosphorylated and active forms (p-AXL and p-MET), are presented as western blot results from MPM cell lines. Thank you for your suggestion, we hope our results are now clearer for the reader.
7) Add the experimental procedure of Western blot to the "Materials and Methods".
As correctly suggested by the referee, we added the indicated procedure in the Materials and Methos section, as follows:
“Western Blot Analysis
Protein lysates were obtained by homogenization in RIPA lysis buffer (Sigma-Aldrich, MO, USA) with protease and phosphatase inhibitors cocktail (Hoffmann-La Roche). Protein extracts were quantified by using Bradford assay (Bio-Rad, CA, USA) and equal amounts of total protein (40 μg/lane) were separate by 4–15% gradient mini precast TGX gel (Bio-Rad) before transferring to nitrocellulose by standard western conditions, blocked in BSA solution and primary antibodies (1:1000 in BSA solution; incubated overnight at 4 °C). The secondary antibody (1:3000 in 5% milk/TBS/Tween20 solution) was incubated at RT for 1 h before detection. Immunocomplexes were detected with the enhanced chemiluminescence kit ECL plus, by Thermo Fisher Scientific (Rockford,IL) using the ChemiDoc (Bio-Rad). Values were normalized to α-tubulin. Each experiment was done in duplicate.”
8) Add scale bars to all the images.
As correctly observed by the referee, the Figures 1 and 2 were missing the scale bars, we have inserted the correct figures.
9) Make the figure formats the same of Figure 3a and 3b. Make sure the writing is correct, eg, it should be "0.5" other than"0,5".
Thanks for your suggestion, we have unified the format of the figures 3a and 3b and we changed the writing 0,5 in 0.5.
10) Why are the error bar of the CTR in Figure 3c and 3d are the same?The data or results should be wrong.
As correctly observed by the referee, the error bars of the CTR in Figure 3c and 3d are not the same. The experimental design could explain this difference; the two figures show the results of two different experiments of the colony forming abilities of mesothelioma cell lines, particularly one after the treatment with foretinib and the other with cabozantinib.

Reviewer 2 Report
The authors analyzed the co-expression of AXL and MET in a patient cohort of 72 PM (please note a new term used according to WHO guidelines, now PM instead of MPM) using IHC and FISH. Afterwards they tested the efficacy of AXL-MET multi-target inhibitors in 4 PM cell lines. There are a few points in the method section that are unclear:
1. IHC scoring, the authors only applied score when there were more than 10% of tumor cells showing positive or negative staining. Which score then was given for the tissues that were <10% positive?
2. Future tense was used in some parts of the method section, please correct.
3. What was used as normal cell counterpart for FISH analysis and how many of the normal cells were counted?
4. Which software was used to calculate IC50
5. Which system was used for disease staging? (eg. IMIG)
In general, it is an interesting study concept to test multitarget inhibitors as the single inhibitors showed minimal efficacy. But these following points questioned whether the interpretation of the data and the conclusion were valid.
1. The authors only described the data (IHC and FISH) in a text form. But it is not clear because the data from the whole cohort was not shown. They should describe the results in a form of table or figure to show the distribution of the data of both IHC and FISH in the same table/figure. This will help understanding the data better.
2. Only 4 cases with AXL positive showed AXL gene amplification, do these cases express weak or strong AXL IHC?
3. The author concluded that the AXL and MET co-overexpression is a common event in MPM but there are only 10% of patients that showed co-expression, I disagree with this statement.
4. How was the correlation between co-expression and histotype was performed? Can the authors show the graph?
5. The western blot (AXL, MET) result of the cell lines described in 3.5 was missing.
6. The authors wanted to conclude that multitarget inhibitors are better than single target inhibitors, they should also compare the efficacy with AXL or MET single target inhibitors.
7. No data from apoptosis assay (described in the method section) was shown, please add
8. Last line of the result section, figure 3 was the result of colony formation assay and did not suggest invasiveness.
9. Figure 3: why is this the anchorage-independent colony formation assay? The cells were seeded in a 6 well plate and they would attach on the dish, the assay was not done in a soft agar or any matrix.
Author Response
Reviewer 2:
The authors analyzed the co-expression of AXL and MET in a patient cohort of 72 PM (please note a new term used according to WHO guidelines, now PM instead of MPM) using IHC and FISH. Afterwards they tested the efficacy of AXL-MET multi-target inhibitors in 4 PM cell lines. There are a few points in the method section that are unclear:
- IHC scoring, the authors only applied score when there were more than 10% of tumor cells showing positive or negative staining. Which score then was given for the tissues that were <10% positive?
As correctly observed by the referee, the IHC scoring was unclear about the cases showing the staining in <10% of tumor cells. Thank you for the suggestion. In this view, we modified the main text in the Materials and Methos section, as follows: “no staining or weak staining in <10% of tumor cells, score 0;”
- Future tense was used in some parts of the method section, please correct.
Thanks for your observation, we have revised the main text. All changes are marked in the main text.
- What was used as normal cell counterpart for FISH analysis and how many of the normal cells were counted?
We have used the mesothelium as normal cell counterpart for FISH analysis, we counted the signals in approximately 20 normal cells available in each sample. Thank you for your suggestion. In this view, we modified the main text in the Materials and Methos section, as follows: “The cells of the mesothelium were used as normal counterpart for FISH evaluation.”
- Which software was used to calculate IC50
The IC50 was determined by interpolation from the dose–response curves. Results represent the median of 3 separate experiments, each performed in quadruplicate.
- Which system was used for disease staging? (eg. IMIG)
As correctly observed by the referee, the system used for disease staging was missed in the main text. We have used the eighth TNM classification for malignant pleural mesothelioma to define the stage of the cases analyzed in our series, however we had forgotten to specify it in the text. Thank you for your suggestion that has improved our manuscript.
In this view, we have added the reference “19. Berzenji, L.; Van Schil, P.E.; Carp, L. The eighth TNM classification for malignant pleural mesothelioma. Transl. Lung Cancer Res. 2018,7:543-549. https://doi.org/10.21037/tlcr.2018.07.05” and modified the main text in the "materials and methods 2.1. Patients Cohort” section as follows:
“The disease stage was defined according to the eighth TNM classification for malignant pleural mesothelioma [19].”
In general, it is an interesting study concept to test multitarget inhibitors as the single inhibitors showed minimal efficacy. But these following points questioned whether the interpretation of the data and the conclusion were valid.
- The authors only described the data (IHC and FISH) in a text form. But it is not clear because the data from the whole cohort was not shown. They should describe the results in a form of table or figure to show the distribution of the data of both IHC and FISH in the same table/figure. This will help understanding the data better.
As correctly suggested by the referee, we have implemented our results with a table that summarizes the AXL and MET (both IHC and FISH results) associated with all the clinical-pathological characteristics of the series analyzed. Moreover, we have described the results more accurately, all changes are marked in the main text. We thank the reviewer for his suggestions which made the manuscript much clearer.
- Only 4 cases with AXL positive showed AXL gene amplification, do these cases express weak or strong AXL IHC?
As correctly observed by the referee, we missed the description of the scores of the positive cases. Thus, we have implemented our results for AXL IHC and MET IHC. Particularly, regarding the cases with AXL IHC positive results and AXL gene amplification, we changed the sentence as follows: “Among 10 cases with AXL gene amplification, only 4 cases showed AXL expression (3 cases were score 3 and 1 case was score 1) suggesting not a close association between the activation of the receptor and the gene amplification.”
- The author concluded that the AXL and MET co-overexpression is a common event in MPM but there are only 10% of patients that showed co-expression, I disagree with this statement.
As correctly suggested by the referee, our statement is too excessive in relation to the percentage of cases carrying AXL-MET co-expression observed in our series. Our findings suggest that AXL and MET co-overexpression could represent a potential mechanism of cooperation of these RTKs in the MPM pathogenesis, however only a small subset of MPMs may be affected. In conclusion, our sentence proposed in the Discussion was really confusing, thus we modified it as follows: “Our results in FFPE specimens showed that AXL and MET can be simultaneously expressed in MPM suggesting a possible cooperation of these RTKs in the pathogenesis of this neoplasia.” We thank the reviewer for his suggestion.
- How was the correlation between co-expression and histotype was performed? Can the authors show the graph?
We have performed the Pearson χ 2 test to determine whether a relationship exists between the clinical-pathological features included in our study and AXL-MET expression. This statistical test was conducted using SPSS 20.0 for Mac (SPSS Inc., Chicago, Ill). The output of the statistical results by SPSS is summarized in a crosstabulation, therefore we cannot provide a graph. Moreover, we have made some changes in the main text in the "materials and methods" section to make the statistical method used more readable.
- The western blot (AXL, MET) result of the cell lines described in 3.5 was missing.
We have performed the western blot analysis to evaluate the expression levels of total and phosphorylated isoforms of AXL and MET, however we had not inserted the related images in the main text. As correctly suggested by the referee, we added the western blot results in Figure 3. Thank you for your suggestion, we hope our results are now clearer for the reader.
- The authors wanted to conclude that multitarget inhibitors are better than single target inhibitors, they should also compare the efficacy with AXL or MET single target inhibitors.
As correctly observed by the referee, our study lacks in vitro studies related to the efficacy of the AXL or MET single target inhibitors. This aspect has already been extensively treated in previous in vitro studies (for example Ou, WB et al, Oncogene. 2011; Pinato, DJ et al, Br. J. Cancer. 2013), moreover also clinical data have been showed that the single-agent RTK specific inhibition is clinically ineffective in patients with MPM, despite other cancer types. In this context, the main aim of the present study was rather investigating the efficacy of the multitarget inhibitors, including AXL and MET as targets, as novel therapeutic approach to be further investigated in MPM.
- No data from apoptosis assay (described in the method section) was shown, please add.
We apologize for the mistake, we have incorrectly entered in the Materials and Methods the section “Assessment of apoptosis”. The apoptosis experiments were not performed in our study, thus we have eliminated the paragraph 2.4.4 from the main text.
- Last line of the result section, figure 3 was the result of colony formation assay and did not suggest invasiveness.
Thanks for your comment. We used the colony formation assay, also known as clonogenic assay, as an in vitro cell survival assay based on the ability of single cells to grow into colonies. We present it as a test of cell survival after treatment; in particular, the small fraction of seeded cells that retains the capacity to produce colonies is to be considered as a subgroup of aggressive and potentially invasive cells. However, we revised the text according your suggestion.
- Figure 3: why is this the anchorage-independent colony formation assay? The cells were seeded in a 6 well plate and they would attach on the dish, the assay was not done in a soft agar or any matrix.
Thank you, we revised the text and we present the results as clonogenicity / colony forming assay: cells need to be seeded at very low densities and left for a period of 1-3 weeks for colonies to form. Colonies are then fixed, stained with crystal violet to make them visible, and counted.The assay measures the efficiency by which cells form colony units when plated at clonogenic levels in monolayer culture on tissue culture plastic.

Round 2
Reviewer 1 Report
Thanks for addressing most of my concerns. However, as a key word in the title "co-expression", I still think the authors should do a co-staining of AXL and MET in patient samples and cultured cells, and even after drug treatment. According to my extensive experience, I do not think the authors' answer to my question is true "Unfortunately, we are unable to provide the co-staining images of the two receptors since they have the same cellular localization.". I still do not know why you can not do it when they are in the same cellular localization. Many experiments and papers have been done in this way. I have done the experiment for three proteins with the same localization (nucleus), and there was no problem when you using well separated secondary fluorescent antibody. Then, you can easily check their experession and localization under confocal microscopy.
Author Response
Dear Reviewer,
thank you for appreciable suggestions that greatly improved our manuscript.
In addition we performed a double staining through immunofluorescence for Axl and Met in both cell line H2452 and in mesothelioma coexpressing cases. Thus, we confirmed though this method that the cases co-express both receptors. I think that the manuscript acquired a more effective value.
Thank you again for your support
Cordiality
Federica Zito Marino
Reviewer 2 Report
The authors have addressed all the comments and added more data . I think the manuscript has sufficient quality for publication. Nevertheless, there are still some spelling mistakes and need some proofreading.
Author Response
Thank you for your support